# Decipher the Immunopathological Mechanisms and Set Up Potential Therapeutic Strategies for Patients with Lupus Nephritis

**DOI:** 10.3390/ijms241210066

**Published:** 2023-06-13

**Authors:** Chang-Youh Tsai, Ko-Jen Li, Chieh-Yu Shen, Cheng-Hsun Lu, Hui-Ting Lee, Tsai-Hung Wu, Yee-Yung Ng, Yen-Po Tsao, Song-Chou Hsieh, Chia-Li Yu

**Affiliations:** 1Division of Immunology & Rheumatology, Department of Medicine, Fu Jen Catholic University Hospital & College of Medicine, Fu Jen Catholic University, New Taipei City 24352, Taiwan; 2Division of Rheumatology, Immunology & Allergy, Department of Internal Medicine, National Taiwan University Hospital, National Taiwan University College of Medicine, Taipei 106319, Taiwan; dtmed170@gmail.com (K.-J.L.); tsichhl@gmail.com (C.-Y.S.); b98401085@ntu.edu.tw (C.-H.L.); hsiehsc@ntu.edu.tw (S.-C.H.); 3MacKay Memorial Hospital & MacKay Medical College, New Taipei City 25245, Taiwan; htlee1228@gmail.com; 4Division of Nephrology, Department of Medicine, Taipei Veterans General Hospital and Faculty of Medicine, National Yang-Ming Chiao-Tung University, Taipei 112304, Taiwan; thwu@vghtpe.gov.tw; 5Department of Medicine, Fu Jen Catholic University Hospital & College of Medicine, Fu Jen Catholic University, New Taipei City 24352, Taiwan; yyngscwu12@gmail.com; 6Division of Holistic and Multidisciplinary Medicine, Department of Medicine, Taipei Veterans General Hospital and Faculty of Medicine, National Yang-Ming Chiao-Tung University, Taipei 112304, Taiwan; dolphinandy@gmail.com

**Keywords:** lupus nephritis, anti-dsDNA antibodies, cross-reactivity, renal resident cells, NLRP3 inflammasome, type I interferon

## Abstract

Lupus nephritis (LN) is one of the most severe complications in patients with systemic lupus erythematosus (SLE). Traditionally, LN is regarded as an immune complex (IC) deposition disease led by dsDNA–anti-dsDNA-complement interactions in the subendothelial and/or subepithelial basement membrane of glomeruli to cause inflammation. The activated complements in the IC act as chemoattractants to chemically attract both innate and adaptive immune cells to the kidney tissues, causing inflammatory reactions. However, recent investigations have unveiled that not only the infiltrating immune-related cells, but resident kidney cells, including glomerular mesangial cells, podocytes, macrophage-like cells, tubular epithelial cells and endothelial cells, may also actively participate in the inflammatory and immunological reactions in the kidney. Furthermore, the adaptive immune cells that are infiltrated are genetically restricted to autoimmune predilection. The autoantibodies commonly found in SLE, including anti-dsDNA, are cross-reacting with not only a broad spectrum of chromatin substances, but also extracellular matrix components, including α-actinin, annexin II, laminin, collagen III and IV, and heparan sulfate proteoglycan. Besides, the glycosylation on the Fab portion of IgG anti-dsDNA antibodies can also affect the pathogenic properties of the autoantibodies in that α-2,6-sialylation alleviates, whereas fucosylation aggravates their nephritogenic activity. Some of the coexisting autoantibodies, including anti-cardiolipin, anti-C1q, anti-ribosomal P autoantibodies, may also enhance the pathogenic role of anti-dsDNA antibodies. In clinical practice, the identification of useful biomarkers for diagnosing, monitoring, and following up on LN is quite important for its treatments. The development of a more specific therapeutic strategy to target the pathogenic factors of LN is also critical. We will discuss these issues in detail in the present article.

## 1. Introduction

Systemic lupus erythematosus (SLE) is a chronic systemic autoimmune disorder with multifactorial etiopathogenesis and a female preponderance. Lupus nephritis (LN) is a common and rather severe complication in patients with SLE. Approximately 30–50% of adults and up to 60–70% of children with SLE suffered from LN within 5 years after diagnosis [1]. Despite of intensive treatments, LN could still progress to end-stage renal disease (ESRD) in 5–10% [2]. The diagnosis of LN relies on the renal biopsy based on the 2003 International Society of Nephrology (ISN) Pathological Classes [3]. The spectrum of its clinical manifestations ranges from silent urinary abnormalities to highly overt nephritic syndrome or even rapid progression to renal failure, as judged from the biopsy findings [4]. The incidence of LN varies with ethnicities [5] and becomes one of the major causes of death in SLE patients [6]. In addition to the glomerular injury as the primary lesions in LN, increasing attention has been focused on the interstitial fibrosis, renal tubular atrophy, and vascular lesions as markers of severity in renal injury [3,7]. Besides, the importance of chronicity index and its role in predicting treatment response and renal prognosis (as represented by renal fibrosis) have caught much attention [8]. Obviously, LN is one of the most common causes of death in patients with SLE, along with infection, cardiovascular events, and malignancies [6]. The progression of LN to chronic kidney disease and end-stage renal outcome may originate from renal fibrosis. The molecular mechanism for extravascular matrix (ECM) accumulation leading to tissue fibrosis is worthy for further exploration.

Recently, large numbers of new biologics targeting B cells, T cells, cytokines, inflammatory factors, non-coding RNAs, or signaling pathways have emerged and may become potential therapeutic modalities. They have been extensively reviewed [9,10]. For further updating the novel conception for LN as a whole, we’ll discuss these points in detail in the following sections.

## 2. Recent Advances in the Pathophysiologic Mechanisms for LN

### 2.1. Selective Accumulation of Adaptive Immune Cells in LN

Conventional T [11,12] and B [13] lymphocytes have been found to be the major infiltrative immune cells in LN, sometimes with well-organized aggregates similar with a secondary lymphoid structure. Yamada et al. [14] is the first group to examine the expression of surface markers, CCR4 and CCR5, in peripheral blood lymphocytes (PBLs) and mononuclear cells infiltrating into the renal tissue of LN patients. They found that CCR4^+^ T lymphocytes (F_h2_ subpopulation) migrated into the renal tissue of these patients. The CCR4^+^ F_h2_ subpopulation migrating into the renal tissue might be implicated in the pathogenesis of LN. Murata et al. [15] analyzed TCR Vβ1–20 gene families in intra-renal T cells and PBLs by a nested RT-PCR and southern blot. They found the repertoire of TCR Vβ in the intra-renal T cells was relatively restricted. Actually, TCR Vβ8 and Vβ20 genes were preferentially expressed in 50% of intra-renal T cells and TCR γβ9 as well as γβ14 genes were expressed in 42% of intra-renal T cells, much more frequent than those in the PBLs. These oligoclonal expressions were higher and more conspicuous in intra-renal T cells than the expression of IFN-γ mRNA in the same cells, which was negligible.

Crispin et al. [16] reported that expanded double negative T cells in patients with SLE produced IL-17 abundantly in the kidneys. Enghard et al. [17] identified CXCR3^+^ CD4^+^ T cells and found that these cells were enriched in the inflamed kidneys and the voided urine. Recently, Allison et al. [18] demonstrated further that CD8^+^T cell infiltrated in crescentic glomerulonephritis (GN) and caused breaches in Bowman’s capsule. These molecules or cells may become valuable biomarkers for LN activity and may also represent the candidate targets to design future therapies.

### 2.2. Innate Immune Cell Infiltration in LN

Histologically, renal resident macrophages have been shown to play important roles in maintaining local homeostasis and renal organogenesis [19]. However, interstitial accumulation of monocytes/macrophages is a pathological feature associated with a poor prognosis of LN. Single-cell RNA-sequencing has revealed the presence of increased numbers of monocytes/macrophages in the inflamed kidney of SLE patients [20,21,22]. Immunohistochemical studies showed that monocyte/macrophage and/or T cell infiltration(s) in glomeruli and renal interstitium, especially the interstitial macrophages, may be the major effectors for chronic injury correlated with proteinuria and poor renal prognosis in patients with LN [23,24]. Furthermore, Yoshimoto et al. [25] and Nakatani et al. [26] demonstrated that increased glomerular fractalkine expression and CD16^+^ monocyte accumulation was correlated with lupus disease activity and proliferative glomerular LN. Accordingly, the accumulation of infiltrating monocytes/macrophages in the kidney with a pro-inflammatory phenotype correlated with the degree of glomerulosclerosis and interstitial fibrosis by producing inflammatory cytokines and chemokines in LN [27].

It is conceivable that chronic inflammation may lead to prolonged exposure of dendritic cell (DC) to pro-inflammatory cytokines, which subsequently triggers their maturation to aggravate inflammation reciprocally [28]. Scheinecker et al. [29] have found a couple of changes or deficiencies in circulating DCs, in conjunction with an altered reaction to T cell stimulation in SLE patients. Tucci et al. [30] reported that plasmacytoid DCs (*p*DCs) decreased in peripheral blood, but increased or infiltrated in glomeruli in SLE patients, whereas myeloid DCs (mDCs) were almost absent in the same individuals. This particular accumulation of *p*DCs was associated with the production of IL-18, IL-12, and IFN-γ. Similarly, Maria et al. [31] have also demonstrated increased renal macrophages and DCs in LN.

By using bioinformatic analyses, Cao et al. [32] unveiled that five types of immune cells are associated with LN, with monocytes being the most prominent subset. Gene Ontology (GO) functional annotation and Kyoto Encyclopedia of Genes and Genomes (KEGG) analyses indicated that immune responding pathways are significantly enriched in LN. In these studies, Spearman’s correlation statistics have suggested that 15 genes—including *FCERIG*, *CLEC7A*, *MARCO*, *CLEC7A*, *PSMB9,* and *PSMB8—*are closely related to clinical features. The accumulated innate and adaptive immune-related cells exhibiting their pathological effects in kidney tissues to cause LN is illustrated in Figure 1.

### 2.3. Renal Resident Cell Activation in Causing Podocytopathy and Mesangial Proliferation

#### 2.3.1. Podocytes and Lupus Podocytopathy

Podocytes are highly specialized cells residing on the visceral side of Bowman’s capsule and wrapping around glomerular capillaries. These epithelial cells are crucial components of the glomerular filtration barrier by expressing biomarkers—such as synaptopodin, nephrin, podocin, and Wilm’s tumor protein—which have been extensively reviewed by Perico et al. [33]. Autophagy is a highly conserved process that can degrade certain intracellular contents in both physiological and pathological conditions [34,35]. Defects in autophagy contribute to the pathogenesis of SLE [36]. Thus, halting this activity during the early stages of LN may be designed as a potential therapeutic strategy for delaying the progression of LN. Jin et al. [37] demonstrated that differentiated mouse podocytes in the LN group showed a reduced nephrin expression and an increased apoptosis. The study has suggested that autophagic activity of podocytes is a crucial factor for renal damage by directly affecting podocyte function.

Lupus podocytopathy is a glomerular lesion in SLE characterized by a diffuse epithelial cell foot process effacement without an immune complex deposition or with only a mesangial immune complex precipitation. This pattern is a unique form of LN that mimics minimal change or primary focal segmental glomerulosclerosis, as reviewed by Oliva-Damaso et al. [38].

Figure 2 illustrates the physiological functions of podocyte foot processes and the unique features of podocytopathy in LN.

#### 2.3.2. Mesangial Cells (MC) in LN

It has been recognized that MCs play crucial roles in renal physiology and immunology. Their functions may include the production of an extracellular matrix, the repair process after damage, an engulf of the proteins trapped in glomerular basement membrane, communications with other glomerular cell components, and maintenance of glomerular blood flow, as extensively reviewed by Avraham et al. [39] and Liu et al. [40]. Accordingly, renal mesangium becomes one of the primary sites for the immune complex deposition. Besides, MCs constantly undergo damages, resulting in excessive proliferation and extracellular matrix (ECM) production [40]. MC proliferation is a key finding in LN, resulting in a rapid production of inflammatory cytokines, such as type 1 interferon (IFN-I), IL-1β, and IL-6, after Toll-like receptor (TLR) stimulation [41,42].

In real world scenarios, most clinical trials of biologics targeting immune cells or immune mediators, rather than targeting renal resident cells, seem unable to show efficacy in patients with LN. Ever-increasing evidence have indicated the involvement of renal resident cells—including podocytes, mesangial cells, and/or even epithelial and endothelial cells—in the pathogenesis of LN, as reviewed by Kwok et al. [43]. It is expected that these renal resident cells may become the targets or facilitators of novel biologics to treat LN. Figure 3 shows the physiological and pathological roles of glomerular mesangial cells (GMCs) in LN.

### 2.4. Intricate Interactions between Infiltrating Immune Cells and Renal Resident Cells in Causing Kidney Inflammation

In spite of the previous intensive investigations on the pathogenesis of LN, the exact factors that are implicated in its development remain unclear. The complexity of LN relies on the facts that interactions among infiltrated immune-related cells, in situ resident cells, and systemic autoimmunity are all implicated in its immunopathogenesis. It appears that any single factor—including autoantibodies, cytokines, immune complexes, activated complements, or the infiltrated immune cells *per se*—is unable to elicit enough renal damage unless collaborating with renal resident cells—including GMCs, podocytes, pericytes, and renal tubular epithelial cells—as described in the previous section.

Kolios et al. [44] have reported that UV light exposure may cause polymorphonuclear neutrophil (PMN) activation in skin and trigger the so-called “skin–kidney crosstalk” characterized by their migration to the kidney to elicit inflammation. Sung and Fu [45] used a novel technology to observe cytokine signals in the kidney of lupus-prone NZM2823 mice by confocal microscopy. They found that cytokine production within glomeruli was cell-type-specific and under translational control. IL-6, IL-1β, and TNF-α in the inflamed glomeruli were produced predominantly by GMCs, podocytes, and the infiltrated blood-derived macrophages. Evidence supports that these three cell types form a cytokine circuit to amplify cytokine responses in LN. In addition, these intrinsic renal cells and infiltrative mononuclear cells can produce other cytokines—such as M-CSF, SCF, and IL-34—that are present in the enclosed glomerular space to constitute the soluble effector milieu for mediating cellular damage and proliferation, as have been extensively reviewed by Bhargava et al. [46]. More interestingly, the production of IL-10 and IL-1β by infiltrative macrophages was minimal, in spite of a high mRNA expression. However, the macrophages released from Bowman’s capsules could produce large amounts of IL-10 and IL-1β. These data may suggest the complexity of cytokine regulation and production in LN and provide a potential therapeutic strategy for cytokine-blocking therapy in LN.

Tertiary lymphoid structures (TLS) are frequently found in autoimmune diseases, such as rheumatoid arthritis, Sjögren’s syndrome, multiple sclerosis, Hashimoto’s thyroiditis, diabetes mellitus, primary sclerosing cholangitis, and primary biliary cirrhosis. TLS’ with true germinal centers are also present in the kidneys of patients with LN in association with a poor renal prognosis [47]. It is conceivable that renal stromal cells, tubular epithelial cells, high endothelial, and lymphatic vessel cells produce chemokines that enable and facilitate the TLS formation. This structure is composed of T-cell-rich zones with mature DCs next to B cell follicles with a germinal center surrounded by plasma cells [47]. Pathologically, TLS’ may generate autoreactive T and B cells in situ and produce autoantibodies, as well as cytokines, to perpetuate the pathological processes in LN. The majority of the cells in TLS’ are CD3^+^ T cells, including CD8^+^ T and CD4^+^ T cells displaying a Th1 phenotype, and CD4^+^Treg cells [48]. On the other hand, B cells in TLS’ have been demonstrated to be undergoing somatic hypermutation [13]. This would lead to the local production of autoantibodies and a possible immune complex formation in situ. Th17 cell populations have also been shown in the lupus kidney, indicating an inflammatory response to the tissue damage [16,49,50,51]. Besides, the TCR repertoire of renal infiltrating cells in LN is restricted, suggesting kidney-specific antigens are explicitly recognized [52]. It is very likely that T cells destroy kidney resident cells, such as podocytes, through direct cytotoxicity [53] or through the indirect action of cytokines, including IL-23 [54].

Figure 4 depicts the immunopathological injury mediated by systemic autoimmunity, infiltrating immune-related cells and activated resident cells after TLS formation that constitute a vicious cycle to perpetuate LN.

### 2.5. Roles of Chemokines, Cytokinnes and Inflammation-Related Molecules in the Pathogenesis of LN

For elucidating the pathological roles of cytokines/chemokines in LN, Teramoto et al. [55] performed genome-wide mRNA expression analysis of glomeruli microdissections from MRL/*lpr* lupus-prone mice presenting with mesangial proliferative glomerulonephritis (MPGN) with immune cell infiltration. They identified 565 up-regulated genes encoding chemokines and chemokine receptors, including *CCL3*, *CCL4*, *CCL5*, *CXCL9*, *CXCL10*, *CXCL11*, *CXCL16*, *CCR5*, *CXCR3*, and *CXCR6*. These molecules were derived from accumulated Th1 cells that were up-regulated by IFN-γ. Steinmetz et al. [56] further explored the pathological role of *CXCR3* in an MRL/*lpr* mouse model. They found a crucial role of *CXCR3* in the development of experimental LN by acting through its chemoattraction of the pathological effector T cells into the kidneys.

In addition to the increased chemokine expression for chemo-attracting Th1 cells to infiltrate into the murine lupus kidney, Tshilela et al. [57] explored the profiles of cytokine gene expression in isolated glomeruli from MRL/*lpr* mice by RT-PCR. The authors found that TNF-α was dominantly expressed and significantly correlated with a glomerular damage score in lupus-prone mice. A similar finding of TNF-α expression in the thymus and spleen during the early development of nephritis in MRL*/lpr* mice had also been demonstrated in our more previous investigation [58].

In a human study, Lee et al. [59] investigated the B lymphocyte chemoattractant CXCL13, and its receptor CXCR5, expression in renal tissues of patients with LN. They found CXCR5 and CXCL13 were highly expressed in the renal cortex in LN. Furthermore, to determine the cytokine balance in LN patients, laser microdissection-based analysis of cytokine expression in the kidneys had been carried out by Wang et al. [60]. The T cells infiltrating into the kidneys were obtained and the single cell samples of both glomerular and interstitial cells were captured by laser-microdissection. The authors noted that the magnitude of cytokine expression may depend on the classification of renal pathology.

On the other hand, IL-17 might also play a critical role in SLE development. Many investigations have revealed that the expressions of IL-6, TNF-α, IFN-I, hyaluronan, as well as the IL-17/Th17 axis, is increased in the kidney of patients and mice with an active SLE, which is presumed to contribute to LN pathogenesis [61,62].

Figure 5 summarizes the chemokines/chemokine receptors and cytokines involved in the pathogenesis of murine and human LN, as analyzed by microdissection and genome-wide mRNA expression studies.

#### Role of Receptor-Interacting Serine–Threonine Kinase 3 (RIP-3) and NLRP3 Inflammasome Activation in LN

Recently, the role of the innate immune system in the lupus pathogenesis has attracted much more attention. The innate immune cells express a nucleotide-binding leucine-rich repeat receptor (NOD-like receptor, NLR) with the domain of pattern-recognition receptors (PRRs). These receptors can recognize pathogen-associated molecular patterns (PAMPs) and damage-associated molecular patterns (DAMPs) [63]. NLRP3 (the NLR pyrin domain containing 3) is a subtype of NLR that has been extensively studied in autoimmune diseases. The activation of NLRP3 inflammasome may eventually lead to the production of IL-1β and IL-18 [63,64]. Shigeoka et al. [65] revealed that ischemic reperfusion kidney injury could elicit cell damage with tubular cell necrosis through NLRP3 activation. Lorenz et al. [66] further demonstrated that the infiltrating innate cells (DCs and macrophages) in the kidney and certain renal resident cells (ECs, parietal epithelial cells, tubular epithelial cells and podocytes) express NLRP3 inflammasome that can respond to PAMPs and DAMPs.

It is worthy to note that the association of NLRP3 inflammasome with clinical and laboratory features of SLE has been found especially relevant to LN activity and severity [67,68,69]. Fu et al. [70] used a fluorescence-labeled caspase-1 inhibitor probe to detect NLRP3 inflammasome activation in the podocytes of lupus-prone NZM2328 mice and biopsied renal tissue from LN patients. They found that NLRP3 inflammasome activation in podocytes contributes to the development of proteinuria in LN. On the other hand, the activation of RIP-3, a key molecule in the programmed necrosis (necroptosis) pathway [71], may lead to the activation of an NLRP3 inflammasome and necroptosis. Guo et al. [69] demonstrated that RIP3 expression was indeed accompanied by NLRP3 inflammasome activation. The authors observed the activation of RIP3 in podocytes, resulting in the development of LN in NEM2328 mice. Besides, Lv et al. [72] disclosed that CD36—a member of the scavenger receptor class B (SR-B)—and NLRP3 were upregulated in podocytes of LN patients, as well as in MRP/*lpr* mice with renal impairment. Further investigations revealed that CD36 promoted podocyte injury by activating the NLRP3 inflammasome and suppressing autophagy to cause renal damage in LN. On the contrary, Tsao et al. [73] found that NLRP12, similar with NLRP3 in the same NLRP family encoding NACHT, LRR and PYD domains-containing proteins, was reduced in monocytes from lupus patients and mice, leading to a spontaneous activation of innate immune signaling and hyper-responsiveness to nuclear antigens. They hypothesized that NLRP-12 is an immune checkpoint to repress IFN signature and preclude the progression of LN.

Figure 6 shows the interactions of RIPs and the NLRP3 inflammasome in inducing podocyte necroptosis in LN.

### 2.6. Involvement of Autoantigens and Autoantibodies in LN

#### 2.6.1. Pathogenic Anti-dsDNA Antibodies

SLE is characterized by the production of multiple autoantibodies to result in extensive tissue and organ damage. These autoantibodies cause a progressive immune complex deposition and immune-mediated injury, and ultimately lead to cell proliferation, apoptosis, inflammation, and fibrosis in the kidney. It is recognized that anti-double-stranded DNA antibodies (anti-dsDNA) are the classic diagnostic and prognostic marker autoantibodies in SLE and LN. Pathological anti-dsDNA can target not only against dsDNA, but cross-react with α-actinin, annexin II, laminin, proteoglycan heparan sulfate, collagen III/IV, C1q, ribosomal P, N-methyl-D-aspartate receptors (NMDAR), and other molecules [74,75,76,77,78]. In addition, authors have also reported that anti-dsDNA can bind to many cell types in the renal tissues—including cultured mesangial cells, podocytes, vascular endothelial cells, and proximal tubular epithelial cells—to induce cell proliferation, apoptosis, and enhanced gene/protein expression for inflammation/fibrosis [79,80,81,82,83,84,85,86,87]. It is equally interesting that anti-dsDNA antibodies can recognize diverse DNA structures, such as single-stranded DNA, Z-form DNA, bent or elongated dsDNA, DNA-RNA hybrids, peptide-nucleic acid hybrids, and locked-nucleic acids (LNA) [88,89,90]. Accordingly, anti-dsDNA antibodies can bind to a complex of native dsDNA or modified DNA containing the thymine dimer.

Table 1 demonstrates the diverse cross-reactive binding activities of the pathogenic anti-dsDNA autoantibodies with protein and DNA molecules.

#### 2.6.2. Molecular Mechanisms of Pathogenic Anti-dsDNA Autoantibodies in the Development of LN

LN is a major cause of morbidity and mortality, affecting 50–60% of the SLE population throughout the course of disease. Accumulating evidence indicates that anti-dsDNA antibodies not only form immune complexes, but activate complement systems by way of binding, either directly or indirectly, to the cross-reactive cognate antigens, chromatin substances, extracellular matrix components, or resident renal cells [91]. The binding can trigger down-stream inflammatory or fibrotic signaling pathways.

##### Molecular Basis for the Pathologic Effects of Anti-dsDNA Autoantibodies

Our previous studies have demonstrated that anti-dsDNA purified from the serum of active SLE patients cross-reacted with ribosomal P proteins on the surface of the murine liver, spleen, kidney, and the cultured rodent glomerular mesangial cells (RMCs) [92,93]. In addition, we noted that the cytotoxic effects by anti-dsDNA autoantibodies were originated from a not-yet identified mechanism different from the *Fas*, c-*myc,* or *p53* pathways [80]. Furthermore, we have also demonstrated that an increased IL-6 expression in RMCs may become a biomarker for the immune damage and inflammation exerted by anti-dsDNA in LN [41]. Qing et al. [82] reported that pathognomic anti-dsDNA induced number of transcripts—including CXCL1/KC, LCN2, iNOS, CX3CL1 fractalkine, SERPINA3G, and IκB—via both Fc-dependent and Fc-independent mechanisms. α-actinin is an F-actin-binding protein that is present in cell–cell and cell–matrix contact sites, playing important roles in maintaining cell morphology and motility [94]. Mostoslavsky et al. [75] and then Deocharan et al. [76] identified α-actinin as an essential cross-reactive cognate antigen for nephritogenic anti-dsDNA in murine lupus. Zhao et al. [95] demonstrated that α-actinin is a major cross-reactive target for anti-DNA antibodies contributory to the renal lesions in SLE patients.

Figure 7 illustrates the possible molecular mechanisms of pathogenic anti-dsDNA autoantibodies in cell apoptosis, inflammation, and tissue fibrosis.

##### Involvement of Glycosylation of Anti-dsDNA Autoantibodies in LN and Its Possible Mechanisms

Although anti-dsDNA antibodies have been regarded as the prognostic marker autoantibodies for predicting the outcome of SLE with LN, Riboldi et al. [96] doubted their clinical values in SLE patients. Devey et al. [97] carried out a series studies on the binding affinity of subclasses of IgG anti-dsDNA antibodies obtained from SLE. These authors found that the high-affinity IgG1 and IgG4 anti-dsDNA antibodies shift from blood to kidney. These results may suggest that the IgG isotype of anti-dsDNA antibodies with their variant antigen-binding affinity may determine the pathogenetic capacity. Furthermore, Kaneko et al. [98] unveiled that sialylation of the IgG Fc glycans may determine the anti-inflammatory activity of IgG antibodies. Käsermann et al. [99] analyzed the functional changes of increased Fab-sialylation of intravenous immunoglobulin (IVIG) by fractionation and found that this sialylation enhanced immunosuppression. Otani et al. [100] revealed that sialylation of IgG3 cryoglobulins determined their nephritogenicity. Recently, Han et al. [101] analyzed the glycosylation—including sialylation, galactosylation, and fucosylation—of IgG anti-dsDNA and identified their IgG subclasses in SLE patients accordingly to the SLE disease activity index (SLEDAI). The group found that glycosylation of the IgG anti-dsDNA rendered these autoantibodies different from the total IgG. Fucosylation of IgG1 anti-dsDNA correlated well with lupus disease activity. Liou et al. [102] disclosed that sialylated IgG anti-dsDNA antibodies mitigate, whereas de-sialylated IgG counterparts worsen the pristane-induced female BALB/c lupus mice. To explore the molecular basis for alleviating SLEDAI by sialylated IgG anti-DNA, the same group [103] further demonstrated that α-2,6-sialic acid (SIA)/IgG anti-dsDNA ratios inversely correlated with the SLEDAI score by enhancing the B cells’ IL-10 production and SIA/IgG anti-dsDNA ratios.

Table 2 depicts the different effects of glycosylated anti-dsDNA on LN.

#### 2.6.3. The Autoantibodies Co-Existing with Anti-dsDNA That Are Involved in LN

In addition to anti-dsDNA antibodies, many other autoantibodies may also participate in the pathogenesis of LN either independently or in association with anti-dsDNA antibodies.

Tsai et al. [104] firstly reported that polyclonal anti-cardiolipin antibodies (aCL) purified from the sera of patients with active SLE could bind to cultured RMC to suppress cell proliferation and induce apoptosis.

Oelzner et al. [105] found both anti-C1q and anti-dsDNA antibodies were detectable in 74% of active LE patients. None of them were negative for both anti-C1q and anti-dsDNA antibodies whereas 37% of SLE patients without active LN were negative for both antibodies (*p* < 0.01). These results were positive findings favoring that both anti-dsDNA and anti-C1q Abs are highly specific for triggering active nephritis. Later, Kianmehr et al. [106] also confirmed that the levels of anti-C1q and anti-dsDNA antibodies are positively correlated to the severity of LN in SLE patients. The prevalence of these autoantibodies was associated with the severity of LN biopsies.

In murine SLE, α-actinin has been identified as a major cross-reactive molecule for the pathogenic anti-dsDNA antibodies [74,75,76,77,78]. In human SLE, anti-α-actinin autoantibodies were also associated with anti-dsDNA. Renaudineau et al. [107] concluded that anti-α-actinin reactivity was associated with high avidity anti-dsDNA antibodies, which are related to the actin-binding site of α-actinin. These findings have suggested that the detection of anti-actinin antibodies is equally unique to that of anti-dsDNA antibodies and anti-α-actinin may become a new autoantibody biomarker for LN.

Onishi et al. [108] screened autoantigens that could react with LN patient sera by using an N-terminal biotinylated protein library from a wheat cell-free protein production system (17 proteins). They noted that 2 LN-associated autoantigens, ribosomal RNA-processing protein 8 (RRP8) and spermatid nuclear transition protein 1 (TNP1), were identified by immunoprecipitation and immunofluorescence in renal tissues. Furthermore, both proteins are so cationic that their respective cognate antibodies could not cross-react with negatively charged dsDNA. Besides, circulating anti-RRP8 and anti-TNP1 autoantibodies were found deposited as immune complexes in glomeruli of patients with LN. Accordingly, these unique autoantibodies may be useful for the prediction of LN in SLE subsets negative for anti-dsDNA antibodies.

The clinical significance of co-existence of anti-dsDNA and anti-ribosomal P protein antibodies (anti-ribosomal P) in LN patients remains debated. Sarfaraz et al. [109] argued that anti-ribosomal P is highly specific for SLE patients with a negative ANA. The co-existence of anti-ribosomal P with anti-dsDNA may offer a protective role in LN. In contrast, Wakamatsu et al. [110] found the co-existence of anti-ribosomal P and anti-dsDNA antibodies in SLE patients favored proliferative nephritis and was associated with prolonged hypocomplementemia and CKD.

Interestingly, Shang et al. [111] demonstrated the serum levels of four autoantibodies—including anti-dsDNA, anti-nucleosome, anti-C1q, and anti-histone antibodies—were significantly higher in active LN with moderate-to-severe disease activity.

Table 3 demonstrates the co-existence of certain autoantibodies and anti-dsDNA with their clinical significance.

## 3. Current Useful Biomarkers in the Renal Tissues of Patients with LN

Traditional laboratory biomarkers, including serological tests, are well-established tools for clinical evaluation of LN, such as anti-dsDNA antibodies and complement C3/C4 levels. Nevertheless, these clinically useful serological changes cannot simultaneously reflect LN flare and discriminate between an active ongoing disease and chronic tissue damage, as has been reviewed by Soliman et al. [112]. Owing to the recent technological progress, such as genomic, proteomic, and metabolomic analyses, searching for novel useful biomarkers in biopsied renal tissues or urine has been advocated, as has been extensively reviewed by different authors [113,114,115].

### 3.1. Immune Gene Expression in Renal Tissue as Disease Activity Biomarkers for LN

#### 3.1.1. Expression of BAFF, APRIL and Their Corresponding Receptors as Disease Activity Biomarkers for LN

B cell maturation/differentiation is controlled by the BAFF system, composed of B cell Activating Factor of tumor necrosis Factor superfamily (BAFF); A proliferation-inducing ligand (APRIL) and their receptors; BAFF receptors (BR3); transmembrane activator and calcium modulator and cytophilin ligand interacters (TACI); and B cell maturation antigens (BCMA) [116]. Elevated levels of BAFF have been found to be correlated with autoimmune diseases in humans, mice, and dogs [117]. Schwarting et al. [118] were the first authors to report BAFF expression in cultured murine and human lupus renal tubular epithelial cells (TEC), correlating to their histopathological activity index. This finding may indicate that renal-derived BAFF plays an important role in lupus nephritis. Later, Suso et al. [119] discovered that the expression pattern of BAFF and its receptors (BAFF-R and TACI) differed according to LN class and could be used as a routine biomarker in LN biopsies. Aguirre-Valencia et al. [120] evaluated the relationships between LN activity and urinary expression levels of BAFF, APRIL, and their respective receptors. The group concluded that urinary mRNAs encoding BAFF/APRIL and TACI/BCMA/BR3 may become useful biomarkers for LN. In addition, Marín-Rosales et al. [121] confirmed that the expression of BAFF and BAFF receptors in renal tissue is mainly associated with class IV LN and are probably essential pathogenic factors for LN.

#### 3.1.2. Expression of IFN-α and TNF-α Genes in Renal Tissues as the Biomarkers for Disease Activity in LN

Mejia-Vilet et al. [122] compared the differences in glomerular and tubulointerstitial immune gene expression between flared and de novo LN. The total RNAs were extracted and amplified by RT-PCR from laser microdissection, formalin fixed, and paraffin-embedded kidney tissues. They found that a set of eight IFN-α controlled genes were expressed significantly higher in the initial diagnostic biopsy than in the flare biopsy. In contrast, the nine TNF-α-controlled genes were conversely higher expressed in flare biopsied tissues. Fairhurst et al. [123] demonstrated that renal resident cells, rather than infiltrating immune cells, were the major sources of IFN-α in the LN kidney. Another potential source of nucleic acids in the immune-stimulation site is the PMN-derived NETs in glomeruli and renal tubules [124,125,126]. These nucleic acid molecules may activate pro-inflammatory/immunological reactions, via IFN-α and TNF-α production, to impair renal function in class IV LN.

#### 3.1.3. Immune Gene Expression in the Infiltrating Immune Cells in LN

Wang et al. [127] analyzed gene expression relevant to the macrophages and interferons in kidney tissues and the peripheral blood of LN patients. Their results showed that many pathways—including cell cycle, cytoplasmic DNA sensing, NOD-like receptor signaling, proteasome, and RIG-1 like receptors (RLRs)—were activated in peripheral blood. In contrast, the infiltrating immune cells in the kidney showed an arousal of genes relevant to CD4^+^ T cells, CD8^+^ T cells, and DCs, which are strongly implicated in the development and progression of LN.

Immune gene transcript profiling was also performed by Gilmore et al. [128] using the NanoString nCounter Platform on RNA in patients with LN. The results showed the highest expression of IFN-I signaling, complement, and MHC-II pathways in class IV LN. The renal biopsied tissues with class IV LN also exhibited significantly higher upregulated NF-κB signaling and immune genes than those with class V LN.

Molecular profiling of the kidney compartments from serial biopsies was conducted by Parikh et al. [129] to try to differentiate treatment responders from non-responders in LN. Cluster analysis revealed that glomerular integrin, PMN, chemokines/cytokines and tubulointerstitial chemokines, T cells, and leukocyte adhesion genes could differentiate non-responders from clinically complete responders. The follow-up biopsied kidney tissues revealed that IFN was only expressed in the glomerular monocytes and extracellular matrix. Moreover, the tubulointerstitial IFN, complement, and T cell transcripts could differentiate non-responders from clinically complete responders.

Table 4 lists the potential biomarkers derived from renal resident cells and the immune-related molecules from the infiltrating immune cells into the kidney.

### 3.2. Potentially Useful Urinary Biomarkers in LN

The ideal urinary biomarkers of LN should be able to predict the early subclinical flares and to follow the response to therapy based on their involvement in the pathogenesis of LN. These biomarkers ought to be characterized by an easy accessibility and capability to reflect the local inflammation, immune responses, and tissue damage. Number of urinary immune-related molecules have been proposed to be the urinary biomarkers for LN, including TGF-β [130], IL-2, IFN-γ [131], MCP-1/CCL2 [132], interferon-γ-inducible protein-1 (IP-10/CXCL10), CXC chemokine receptor 3 (CXCR3) [133], TNF-like weak inducer of apoptosis [134], adhesion molecules, VCAM-1, and *P*-selectin [135]. Yet, none of them have been validated in clinical practice. On the other hand, neutrophil gelatinase-associated lipocalin (NGAL) has been demonstrated to be a potential biomarker to detect early kidney damage [136] and LN [137]. Our group discovered that 24 h urinary excretion of NGAL in SLE patients with renal involvement was correlated to impaired renal function and could become a potential urinary biomarker for renal damage [138,139]. Besides, Wu et al. [140] noted that the Tamm–Horsfall protein (THP) could become a valid disease biomarker for acute and chronic lupus tubulointerstitial diseases. Bahr et al. [141] confirmed that the IL-22 binding protein (IL-22BP) may be used as a non-invasive marker for assessing the disease activity in juvenile SLE and LN.

Bertolo et al. [142] further established a mass cytometric workflow to profile the urinary leukocyte phenotypes in patients with biopsy-proven proliferative LN. They found that a unique subset of activated effector memory T cells expressing CD38 and CD69 and a set of HLA-DR^+^ CD11c^+^ macrophages were present in the urine sediments from LN patients, which could become biomarkers to stratify LN. This also suggests that variations of leukocytic infiltrates are present in the LN kidneys, which may indicate different pathological classifications. Furthermore, single-cell analysis of RNA sequence (ScRNA-seq) technology provides a detailed profile of gene expression at a high resolution, enabling the exploration of the heterogeneity of cell types and the pathological status of LN. On the other hand, urine proteomics and renal signal cell transcriptomics have been carried out by Fava et al. [143]. The group found that IL-16, CD163, and TGF-β reflected intra-renal nephritis activity. ScRNA-seq independently mirrored that IL-16 producing cells were localized at the sites of kidney damage. Lately, analyses of urinary microparticle components as urinary biomarkers of LN have been reported. Burbano et al. [144] found that HMGB1-containing microvesicles in urine could become the hallmark of patients with LN. Lu et al. [145] reported that the increased podocyte-derived annexinV (+) and podocalyxin (+) microparticles were associated with disease activity and renal injury in LN. Gudehithlu et al. [146] noted that increased urinary exosome-derived ceruloplasmin could serve as an early biomarker before the development of overt proteinuria in not only diabetic nephropathy, but also in many non-diabetic chronic kidney diseases, including LN. More interestingly, many different urinary exosome-derived miRs have been found as useful biomarkers in patients with LN. Sole et al. [147,148] reported that the urinary exosomals miR-21, miR-29c, and miR-150 could become predictors of early renal fibrosis in LN. Tangtanatakul et al. [149] disclosed that down-regulated miR-21 and let-7a of urine exosomes in LN patients could be used to guide clinical staging during disease flare-ups. Garcia-Vives et al. [150] demonstrated that miR-135b exhibited the best predictive value to discriminate responder from non-responder for treatments in LN patients. Perez-Hernandez et al. [151] further discovered that urinary exosomal miR-146a was not only serving as a biomarker of albuminemia, correlating with lupus activity, proteinuria, and histological features to discriminate individual LN patients, but also represented as a good baseline marker for SLE flares. Cheng et al. [152] used a bioinformatic screen combined with clinical validation to rapidly extract the exosomal miRs for an LN diagnosis and management. The group screened out differentially expressed miRs and mRNAs in the LN database. They performed an miR–mRNA integrated analysis to select out reliable miRs in LN tissues. The authors have revealed that urinary exosomal miR-195-5p could be used as a novel biomarker in LN. On the other hand, the miR-195-5p–CXCL10 axis has been suggested to become a therapeutic target of LN. Most recently, Chen et al. [153] detected tRNA-derived small noncoding RNA (tsRNA) in urine by exosomic methods. They showed that upregulated tRF-Ile-AAT-1 and tiRNA5-Lys- CTT-1 tsRNAs were present in SLE patients with LN, but not in those without LN. These results may suggest that different urine-derived proteins and microparticle-derived nucleic acids can serve as non-invasive biomarkers for the diagnosis and prediction of SLE patients with LN.

Comprehensive reviews of immune-related urinary biomarkers relevant to the diagnosis, pathogenesis, and prognosis of LN have been published by Aragón et al. [154] and Morell et al. [155]. Table 5 lists the presumed urinary biomarkers in LN.

## 4. Potential Novel Therapeutic Strategy for LN in Future

High-dose corticosteroids in combination with cyclophosphamide or mycophenolic acid analogues (MPAA), followed by a low-dose maintenance of these drugs, are the current standard protocol for induction and maintenance therapy for LN. In addition, the regimens, including B cell repertoire depletion by anti-CD20 and inhibition of B cell survival factor BAFF, by belimumab are also helpful [156,157,158]. However, more specific and personalized approaches have been advocated and are now under investigation. They are targeting the specific pathogenic factors or regulating the post-transcriptional molecules, which could be regarded as a “renaissance of LN therapy”.

### 4.1. Novel Potential Therapeutic Targets or Checkpoints in the Treatment of LN

#### 4.1.1. Antagonists Targeting the CX3CL1 Family

It has been reported that the infiltrating inflammatory leukocytes in the kidneys of immune-mediated renal diseases express CXCR1 [159,160]. Membrane-bound chemokine, fractalkine CXCL1, can be uniquely induced on the endothelial cells by TNF-α and IL-1β [161] to act as a potent chemo-attractant and adhesion molecule for cells expressing its receptor, CX3CR1. Inoue et al. [162] found that a novel potent fractalkine antagonist could delay the development of murine lupus and ameliorate its progression.

#### 4.1.2. NLRP3 Inflammasome and NF-κB Pathway as Targets of Phytochemicals

NLRP3 inflammasome has been reported to be implicated in LN. Its activation can lead to the generation of caspase-1, IL-1β, and IL-18 through the NF-κB signaling pathway [163]. Recently, Su et al. [164] demonstrated that icariin isolated from Chinese medicine Horny Goat weed (Ying Yang Muo) could alleviate murine LN via suppressing the NK-kB activation, production of IL-1β, TNF-α and CCL2, macrophage infiltration, as well as the NLRP3 inflammasome triggering in autoimmune MRL/*lpr* mice. Furthermore, Wu et al. [165] have comprehensively reviewed and summarized that a couple of phytochemicals isolated from variant herbs, fruits, and vegetables—including icariin, sophocarpine, glycyrrhizic acid, phloretin, magnolol, procyanidin B2, curcumin, epigallocatechin-3 gallate, citrate, and baicalein—can block NLRP3 inflammasome pathways in LN.

#### 4.1.3. JAK/STAT Signaling Pathways as Targets

Cai et al. [166] demonstrated that the IL-35-related JAK/STAT signaling pathway is involved in juvenile SLE (JSLE) in a differentially expressed proteins (DEPs) study. Urinary LAIR1, which is upregulated by IL-35 in mesangial cells, was significantly correlated to SLEDAI-2K. The results may suggest that LAIR1 could be a novel potential target of the IL-35-regulated JAK/STAT signaling pathway in JSLE-LN.

Besides, Shi et al. [167] identified that STAT1 and CXCL10 are the key genes in the occurrence and development of LN. IFN-γ can induce CXCL10 expression by activation of the JAK/STAT1 signaling pathway. Triptolide (TPL), an epoxidized diterpene lactone compound purified from *Tripterygium wilfordii* [168], can inhibit CXCL10 via blocking the JAK/SAT1 signaling.

#### 4.1.4. Complement Components as Targets

Sun et al. [169] injected pcDNA-C1q into MRL/*lpr* mice with LN and found that over-expressed C1q decreased the histological damage index in LN mice. In addition, the levels of IL-1β, IL-6, TNF-α, anti-C1q, and anti-dsDNA in renal tissues were also decreased. Moreover, the expression of CD68, Ki67, and NF-κB-related proteins in glomeruli was all attenuated. These results indicated that C1q ameliorated inflammation and macrophage infiltration, as well as mesangial proliferation by inhibiting the NK-κB pathway in renal tissues. Because the abnormal activation of the alternative complement pathway is involved in LN pathogenesis, Chen et al. [170] used the complement factor B, LNP023, to treat MRL/*lpr* mice and evaluated its effectiveness. They found that the treatment not only improved the symptoms/signs, but also regulated the activation of TNF-α, STAT3, PI3K-Akt, HIF, and Erb-B signaling pathways.

#### 4.1.5. Increased Autophagy by mTOR Inhibitors, Rapamycin or Sirolimus, for Cytoprotection of Podocyte Injury

Autophagy is an innate self-destruction process of cells that can either be facilitated or impeded depending on the environmental status surrounding the cell. Mostly recent investigations have revealed that the defective autophagic activity contributes to lupus pathogenesis, especially via adaptive immune responses, as reviewed by Pan et al. [171]. Evidence also indicated that enhanced autophagic activity is essential for the elimination of aged podocytes and maintenance of the podocyte homeostasis, as well as getting rid of the damaged glomerular cells resulted from immune injury [172]. Sirolimus (Rapamicin^®^) is the prototypic agent used to suppress the mechanistic target of Rapamycin (mTOR). Clinically, sirolimus can activate autophagy to prolong the integrity of podocytes and ameliorate the histological lesions by reducing Akt and arousing mTOR in LN [173,174]. However, Qi et al. [175] disclosed that complement-inactivated serum (at 56 °C), IgG’s derived from LN and IFN-α could induce podocyte autophagy, leading to podocyte injury in MRL/*lpr* mice, which indicates that the use of mTOR inhibitors for LN should be very careful.

#### 4.1.6. Type I IFN Signaling Pathway Inhibitors

Enhancer of zeste homolog 2 (EZH2) is a histone–lysine N-methyltransferase mediating di- and tri-methylation of H3K27 (27th lysine in histone 3), which represses gene expression [176], and is an important activator of the type I interferon (IFN-I) signaling pathway. Excessive expression of EZH2 in SLE favors proinflammatory responses [177] via direct methylation of transcription factors—including *p*53, NF-κB, and STAT3 [178]—with subsequent facilitation of B cell differentiation to plasma cells [179]. Wu et al. [180] reported that the EZH2 inhibitor could alleviate LN in a NZB/NZW F1 mouse by constraining the excessive IFN-I signaling pathway to reduce the anti-dsDNA autoantibodies production and prolong its survival.

### 4.2. The Pathogenic and Therapeutic Role of Non-Coding RNAs in LN

Non-coding RNAs act as epigenetic regulators for modulating the protein expression by targeting mRNA without modifying the genetic sequences. Non-coding RNAs are arbitrarily divided into long non-coding RNAs (lncRNAs) and microRNAs (miRs), depending on their length. miRs are a group of RNA with 20–24 nucleotides, whereas lncRNAs are more than 200 nucleotides long. miRs, by binding to the 3′ end of the untranslated region (UTR) of mRNA, can repress translation and/or degrade mRNA, resulting in a decreased specific protein production, as reviewed by Fabian et al. [181]. On the other hand, lncRNAs are a group of RNAs with a length of more than 200 nucleotides containing a linear or circular RNA (circRNA) in which the head and tail of the RNA is connected by a 3′ -> 5′ phosphor-diester band. lncRNAs and circRNAs can wipe up or sponge the effect of miRs, and consequently affect the downstream signaling pathways. Many authors have reviewed the evidence from animal and human studies and suggested that the expression of miRs [182,183] and lncRNAs [184,185] are altered in LN and can become new biomarkers for the disease. Ahmed et al. [186] disclosed an interesting finding that miR-132 and its corresponding lncRNA, SOX2, were reciprocally expressed in the sera of LN patients. They showed an increased miR-132 and decreased lncSOX2 in LN, but SOX2 is more sensitive than miR-132 in the diagnosis of LN.

#### 4.2.1. Molecular Mechanisms of Different miRs in Suppressing LN

Eitner et al. [187] discovered the role of IL-6 in mediating mesangial cell proliferation and matrix formation in vivo. IL-6 and IL-10 can be regarded as biomarkers for SLE, as has been suggested by Chun et al. [188]. Liu et al. [189] disclosed that miR-410 expression in the kidney tissue of MRL/*lpr* mice was decreased compared to that in normal BALB/c mice, whereas the level of IL-6 was overexpressed in MRL/*lpr* mice. Luciferase assay showed that miR-410 bound directly to the 3′-UTR of IL-6 gene, and an overexpression of miR-410 significantly decreased IL-6 production in a rat mesangial cell line, SV40 MES13. Furthermore, miR-410 overexpression significantly decreased TGF-β1 and collagen I/III in SV40 MES13 cells. Conversely, inhibition of miR-410 enhanced IL-6, as well as the fibrosis factor.

Jacob et al. [190] reported that IL-17 plays a priming role in murine LN. Wang et al. [191] further identified that miR-125a-3p is involved in the progression and development of SLE. Zhang et al. [192], in a recent study, found that this miR was down-regulated in patients with SLE and could decrease IL17 levels and suppress renal fibrosis via down-regulating TGF-β1 in LN mice. An in vitro study unveiled that miR-125-3p bound to the 3′-UTR down-regulates the TGF-β1 in SV40MES.

The JAK-STAT signaling pathway is mediated by many cytokines—such as IFN-I, IL-6, and IL-10—that are pertinent in the pathogenesis of LN, as reviewed by Rönnblon et al. [193]. Wu et al. [194] identified hsa-miR-127-3p as a negative regulator for the IFN-I signaling pathway and verified its target in renal mesangial cells. In addition, they found that overexpression of this miR is down-regulated in the renal tissues of LN patients. They further verified that hsa-miR-127-3p targets *JAK*1 through binding to its 3′-UTR and coding region. These results suggest that hsa-miR-127-3p mimics may be used to inhibit JAK1 and IFN-I signaling in LN.

In contrast, the colony stimulating factor-1 (CSF-1) is responsible for macrophage proliferation and differentiation in glomerulus, leading to renal damage [195], and then increases the severity of LN [196]. Liao et al. [197] confirmed the interaction between miR-145 and CSF-1 by dual luciferase reporter assay. In general, miR-145 expression is down-regulated in LN patients and LPS-induced human renal mesangial cells (HRMCs). Theoretically, miR-145 can inhibit inflammatory damage by down-regulating CSF-1 and suppressing LN development in vivo via the CSF1-mediated JAK-STAT signaling pathway.

#### 4.2.2. Molecular Mechanism of Different lncRNAs and circRNAs in the Treatment of LN

lncRNAs (>200 nucleotides in length) are a class of widespread transcriptional outputs as important regulators in many physiological and pathological processes [198]. Liao et al. [199] demonstrated that the linc0949 expression was decreased in SLE-PBMC and correlated closely to SLEDAI-2K and LN. The same group further examined the expression profile of lncRNAs in kidney biopsies from LN patients by RNA sequencing. They found elevated lncRNA RP11-2B6.2 in kidney tissues of LN patients that was positively correlated with the disease activity and IFN scores. Knockdown of lncRNA RP11-2B6.2 in renal cells suppressed the expression of IFN-stimulated genes (ISGs) and phosphorylation of JAK, TYK2, and STAT1 in the IFN-I pathway, but promoted chromosome accessibility and SOCS1 transcription. These results may provide novel therapeutic targets through attenuation of the over-expressed IFN-I signaling in patients with LN.

Circular RNAs are another new class of ncRNAs that act through binding miRNAs to regulate downstream targets. Xia et al. [200] demonstrated that the circELK4 level correlates to LN disease activity. Xu et al. [201] investigated the function of circELK4 in the miR-276-3p/STING/IRF3/IFN-I pathway in murine lupus and patients with LN. Their results have indicated that the cirELK4/miR-27b-3p/STING/IRF3/IRN-I axis may play a crucial role in LN pathogenesis.

#### 4.2.3. Molecular Mechanisms for Other Unique ncRNA Inhibitors in the Treatment of LN

Zhou et al. [202] reported that miR-150 might facilitate renal fibrosis in flared LN patients through down-regulation of SOCS1 in kidney resident cells in vitro. Moreover, higher plasma miR-150 was noted to correlate with faster renal function decline in chronic renal disease patients [203,204]. Liu et al. [205] unveiled that miR-150 promoted renal glomerular mesangial cell aging by targeting Rab1a and Rab31. Luan et al. [206] used locked nucleic acid (LNA)–anti-miR-150 to investigate kidney injury and its corresponding molecular mechanisms in a spontaneous LN mouse model (Fcγr2b−/−). They found that the miR-150 antagonist decreased the elevated renal pro-fibrotic genes (TGF-β1, α-smooth muscle antibody, and fibronectin) and anti-fibrotic gene suppressor of cytokine signal 1 (SOCS1), renal proinflammatory cytokines (IFN-γ, IL-6 and TNF-α), and the infiltrating macrophage number in the kidney. Hao et al. [207] further identified the biological properties and the total number of infiltrating macrophages and other subsets of pro-inflammatory cells to understand the corresponding molecular mechanisms. They concluded that LNA-anti-miR150 can change the proinflammatory M1 and M2 macrophage polarization via the SOCS1/JAK1/STAT1 pathway.

Table 6 summarizes the presumable applications of ncRNAs (miRs, lncRNAs and circRNAs) as novel potential therapeutic strategies for LN in the future.

## 5. Conclusions and Prospects

LN is a rather complicated pathological process implicating the complex interactions among IC deposition, pathogenic autoantibodies, infiltrating immune and inflammatory cells, and the renal resident cells. More interestingly, a lot of miRs, lncRNAs, and circRNAs, which would modulate the expression of different cytokines/chemokines/growth factors, are actively participating in its pathogenesis. Diverse cross-reactive autoantibodies exert cytotoxic effects on renal parenchymal cells and finally lead to renal fibrosis. It is equally interesting that the glycosylation of the Fab portion of these pathogenic autoantibodies can alter their cytotoxic effects in that sialylation alleviates, whereas fucosylation augments the cytotoxicity. Although immunosuppressants and B cell repertoire depletion by monoclonal antibodies against CD20 or B cell growth factors are effective, many authors have tried to use more sophisticated agents to target the aberrant signaling molecules for suppressing the abnormal gene expression. These agents include antagonists or mimics of ncRNAs in treating murine lupus nephritis. It is expected that these novel therapeutic strategies can be applied in treating human LN in the near future. Nevertheless, the following points should be overcome before these potential strategies can be implemented into the clinical applications:(1)The ncRNA mimics or antagonists can be taken by the infiltrated and/or resident cells in the lupus kidneys.(2)Major adverse effects can be avoided.(3)Precision medicine for LN can be established depending on highly specific serum or urinary biomarkers.(4)Personalized regimens for LN should be identified for modulating the individual pathological factor(s).

## Figures and Tables

**Figure 1 ijms-24-10066-f001:**
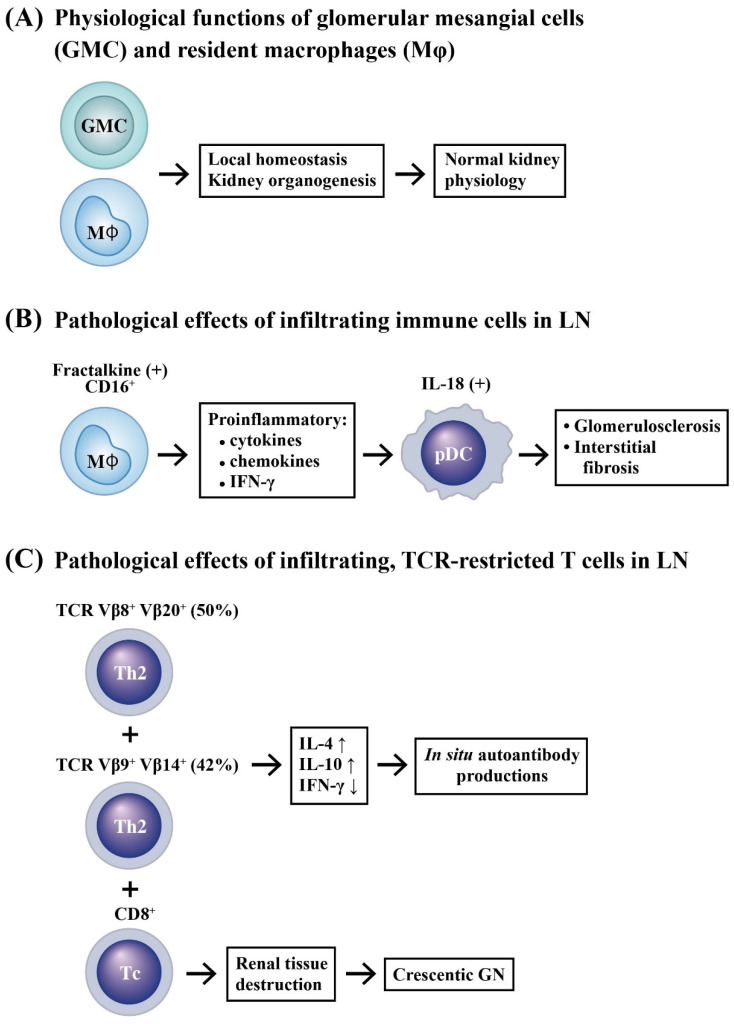
The physiological functions of the resident glomerular mesangial cells (GMCs) and macrophages (Mϕ), and the pathological activities of the infiltrating immune cells in LN. (**A**) The GMCs and resident macrophages can maintain normal kidney physiological functions by keeping local homeostasis and renal organogenesis. (**B**) The specific infiltrating macrophages with the presence of fractalkine and CD16 may produce many different cytokines/chemokines to let dendritic cells (DC) differentiate into IL-18 (+) plasmacytoid DCs *(p*DC), but not myeloid DCs, in the LN kidneys. These changes may lead to glomerulosclerosis and interstitial fibrosis. On the other hand, (**C**) the pathogenic infiltrating T cells with TCR restricted to CCR4^+^ CXCR3^+^ CD4^+^ and bearing TCR Vβ8^+^ Vβ20^+^ (accounting for 50%) or TCR Vβ9^+^Vβ14^+^ (accounting for 42%) produce IL-4 and IL-10 for autoantibody production. In addition, an increased CD8^+^T cell infiltration in the LN kidney facilitates the development of crescentic glomerulonephritis. ↑: Increase in production; ↓: Decrease in production.

**Figure 2 ijms-24-10066-f002:**
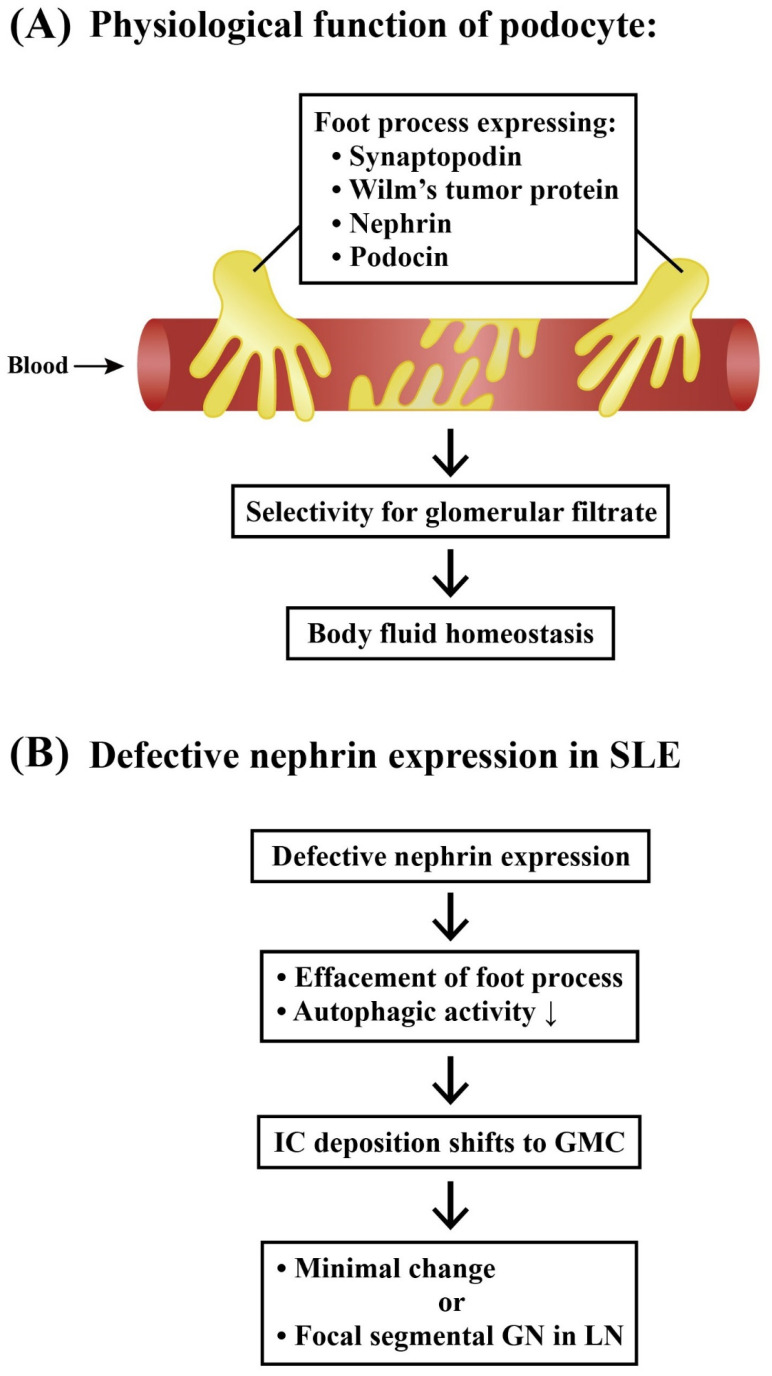
The physiological functions of podocytes and the pathological basis of lupus podocytopathy. (**A**) The normal glomerular podocytes express many peptide molecules on the cell surface of the foot process for selectivity of glomerular filtrates in maintaining the body fluid homeostasis. (**B**) Defective nephrin expression on the podocyte foot process effaces the number of foot processes and suppresses the autophagic activity of podocytes. These changes may diminish the immune complex deposition in a foot process and shift it to GMCs. The shift leads to pathology, simulating a minimal-change disease or primary focal segmental glomerulonephritis in lupus patients showing the typical findings of lupus podocytopathy. ↓: Decrease in production.

**Figure 3 ijms-24-10066-f003:**
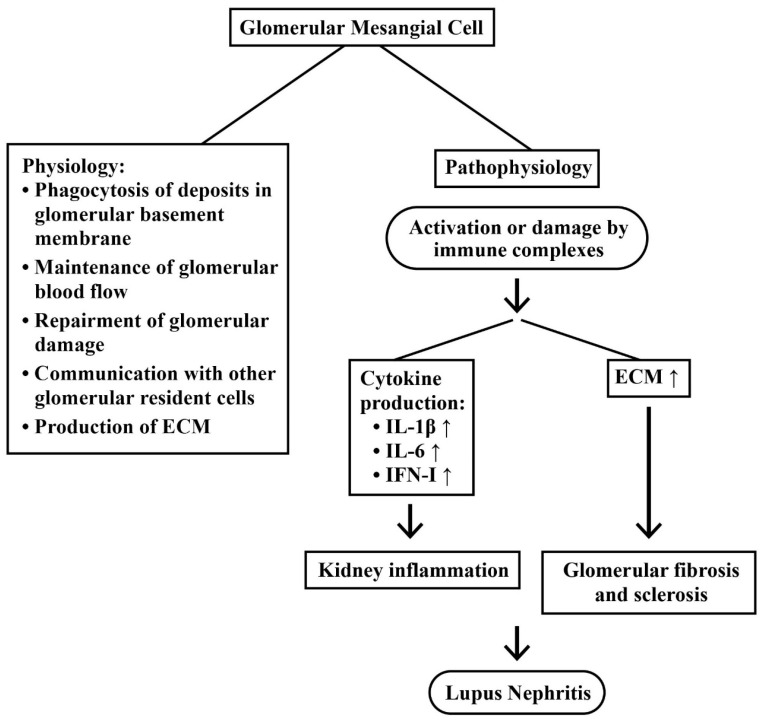
The physiological and pathological functions of glomerular mesangial cells (GMCs) in patients with LN. After activation or damage by the immune complex, a GMC secretes proinflammatory cytokines, IL-1β, IL-6, and IFN-I, to accelerate the inflammation change within the kidney. On the other hand, an extracellular matrix (ECM) is also accumulated to enhance tissue fibrosis. Both of the changes ultimately result in LN. ↑: Increase in production.

**Figure 4 ijms-24-10066-f004:**
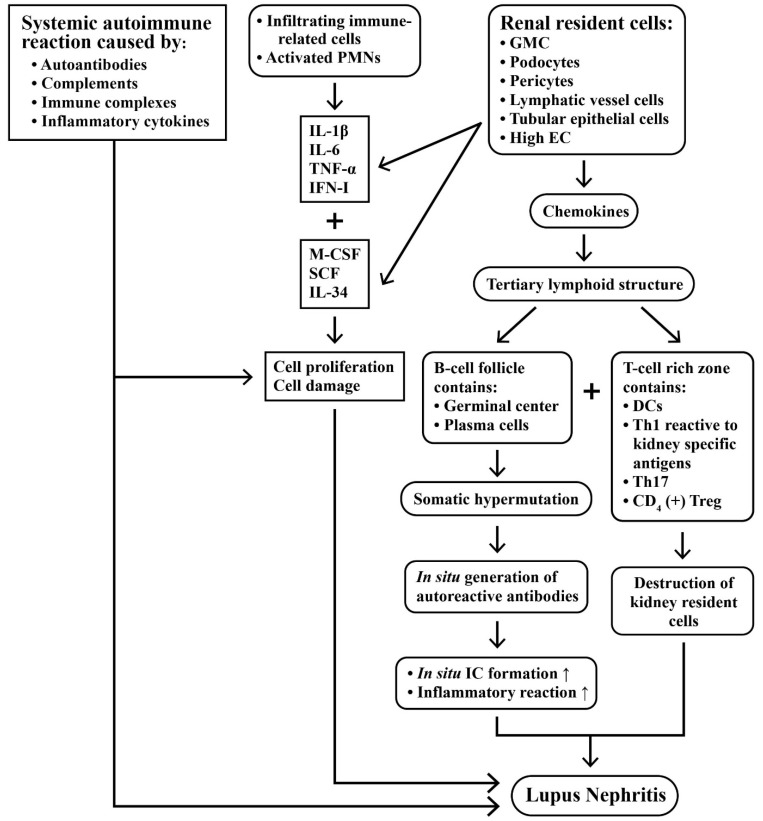
The intricate interactions among a systemic autoimmune reaction, infiltrating immune-related cells and the renal resident cells in inducing lupus nephritis. Not only cytokines, chemokines, and growth factors, but also the formation of a tertiary lymphoid structure (TLS) containing plasma cells with somatic hypermutation is involved in the in situ generation of autoreactive antibodies. In addition, the T-cell-rich zone in a TLS contains Th1 phenotypes, which are reactive with kidney-specific antigens, mature dendritic cells (DCs), Th17 cells, and CD4^+^ Treg to destroy the kidney resident cells. These factors in the tissue altogether induce inflammation, cell proliferation, cell damage, and fibrosis—commonly found in the progression of lupus nephritis. ↑: Increase in production.

**Figure 5 ijms-24-10066-f005:**
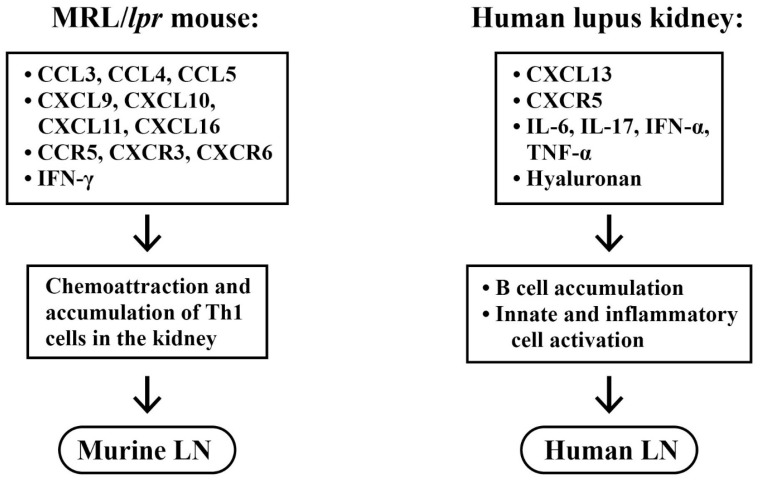
Comparison of different chemokines and cytokines in the induction of murine (**left panel**) and human (**right panel**) lupus nephritis. In human LN, both B cell and inflammatory cell activation accelerate the progression. (Regarding the abbreviations, please refer to the abbreviation list at the end of the text).

**Figure 6 ijms-24-10066-f006:**
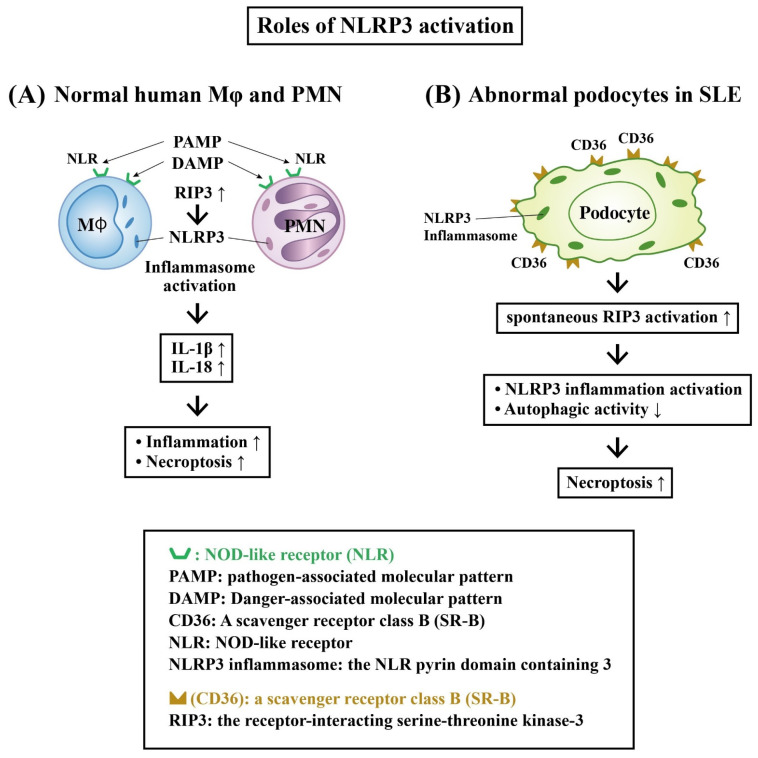
The differences of NLRP3 inflammasome activation between normal human innate immune cells and the renal podocytes in patients with SLE. (**A**) The binding of PAMP or DAMP molecules to NLR activates RIP3 and then NLRP3 inflammasome. The inflammasome activation increases IL-1β and IL-18 production to enhance inflammatory responses and cell necroptosis. (**B**) The podocytes in SLE kidney not only repress NLR, but autonomously express excessive CD36, a class B scavenger receptor, to bind with PAMP and DAMP. This excessive CD36 expression in SLE may facilitate spontaneous RIP3 activation and subsequent NLRP3 inflammasome activation, as well as impeding autophagy activity, and eventually results in necroptosis of podocytes and lupus nephritis. ↑: Increase in production; ↓: Decrease in production.

**Figure 7 ijms-24-10066-f007:**
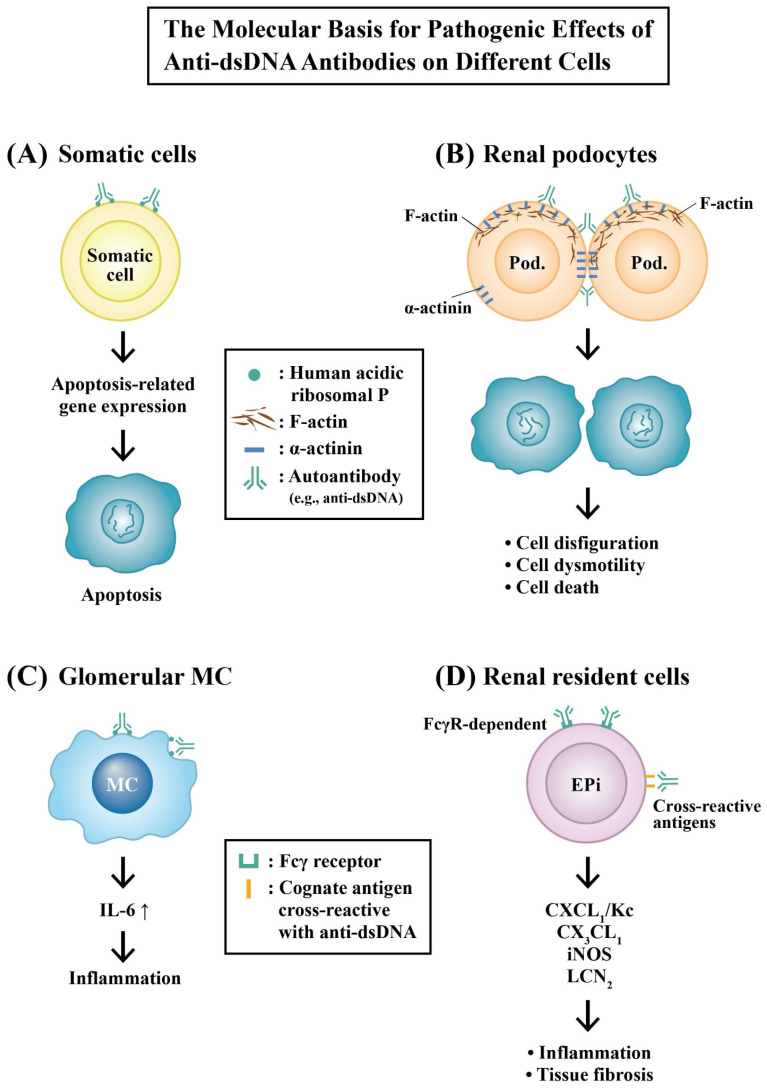
The molecular basis for the effects of pathogenic dsDNA antibodies on somatic and different renal resident cells—including podocytes, mesangial cells, and epithelial cells—by inducing cell damage, inflammation, and tissue fibrosis. (**A**) Anti-dsDNA antibodies cross-react with a human acidic ribosomal P protein expressed on different somatic cells (liver, spleen, or MC) to elicit cell apoptosis. (**B**) Anti-dsDNA antibodies cross-react with α-actinin, an F-actin binding protein present in the cell–cell and cell–matrix contact sites for maintaining cell morphology and integrity. The binding of anti-dsDNA with α-actinin on podocyte cell surface causes disfiguration dysmotility and even cell death. (**C**) The binding of anti-dsDNA with acidic ribosomal P expressed on glomerular mesangial cell surface induce an IL-6 expression and inflammation. (**D**) The anti-dsDNA antibodies may also bind to nonspecific IgG-Fcγ receptors on renal epithelial cells to stimulate excretion of chemokines and other molecules, eventually resulting in chronic inflammation/tissue fibrosis, in addition to the aforementioned cross-reactivity with the surface-expressed molecules. Epi: renal epithelial cell. ↑: Increase in production.

**Table 1 ijms-24-10066-t001:** Different types of DNA and non-DNA molecules present in the renal matrix or on the surface of resident renal cells that can cross-react with pathogenic anti-dsDNA antibodies.

Cross-Reactive with Different Molecules:
• Chromatin substances [88,89,90]:
• dsDNA, ssDNA, Z-form DNA, bent or elongated dsDNA
• DNA-RNA hybrids, peptide-nucleic acid hybrids, locked nucleic acid
• Extracellular matrix components [74,75,76,77,78]:
• α-actinin
• Annexin II
• Laminin
• Heparan sulfate proteoglycan
• Collagen III and IV
• C1q
• Ribosomal P
• N-methyl-D-aspartate receptor
• Others
• Cross-reactive with resident renal cells [79,80,81,82,83,84,85,86,87]:
• Glomerular mesangial cells
• Podocytes
• Vascular endothelial cells
• Proximal tubular epithelial cells

**Table 2 ijms-24-10066-t002:** The effects of isotypes and glycosylation of IgG anti-dsDNA antibodies on their nephritogenic activity.

Sialylation of IgG Fc glycan increases the anti-inflammatory activity of IgG antibodies [98]
Fab-sialylation of IVIG enhances immunosuppression [99]
Sialylation of IgG3 cryoglobulin decreases nephritogenicity [100]
The shift of high-affinity IgG1- and IgG4-isotype anti-dsDNA antibodies from blood to kidney induces LN in SLE [97]
Fucosylation of IgG1 anti-dsDNA antibodies correlates with disease activity in SLE [101]
α-2,6-sialylated IgG anti-dsDNA antibodies suppress LN activity by enhancing IL-10 production from B cells and further increase sialylated IgG antibody production [102,103]

**Table 3 ijms-24-10066-t003:** The clinical significance of the co-existence of anti-dsDNA and other autoantibodies in patients with lupus nephritis.

Co-Existence with Anti-dsDNA Antibodies	Clinical Significance
Anti-cardiolipin antibodies [104]	Induction of GMC apoptosis
Anti-C1q antibodies [105,106]	Active LN
Anti-α actinin [74,75,76,77,78,107]	Biomarker of LN
Anti-ribosomal RNA-processing protein 8 (RRP8) [108]	Prediction for LN in an-dsDNA(-) SLE patients
Anti-spermatid nuclear transition protein 1 (TNP1) [108]	
Anti-ribosomal P antibodies [109,110]	High specific to ANA (-) SLE for protection from LN
	Proliferative LN; Prolongation of hypocomplementemia, and CKD
Anti-dsDNA + anti-nucleosome + Anti-C1q + Anti-histone [111]	Acute LN

**Table 4 ijms-24-10066-t004:** Potential biomarkers expressed in renal tissues and their effects on the infiltrated immune cells correlating with the disease activity of LN.

Potential Biomarkers from Renal Resident Cells	Effect on Infiltrated Immune Cells in the Kidney
BAFF expression in renal tubular epithelial cells [118] BAFF and its receptors (BAFF-R and TACI ↑ [118,120]	IFN-I↑ [128] MHC-II↑ NF-κB↑ Complements↑
8 species of IFN-α mRNA expression↑ [121,122] 9 species of TNF-α RNA↑	Glomerular integrin↑ [129] Chemokines/cytokines↑ T and PMN adhesion molecules↑
PMN-derived NETs↑ [123,124,125]	Extracellular matrix↑ IFN↑ Complement↑

↑: Increase in production.

**Table 5 ijms-24-10066-t005:** The potential urinary biomarkers in the patients with LN.

Urinary BAFF, APRIL and their cognate receptors, TACI/BCMA/BR3 mRNA [120]
Urinary neutrophils gelatinase-associated lipocalin [136,137,138,139]
Tamm-Horsfall protein in tubulointerstitial damage [140]
IL-22-binding protein [141]
CD38^+^ CD69^+^ memory T cell and HLA-DR^+^ CD11c^+^ macrophage in urinary sediments [142]
IL-16, CD163 and TGF-β [143] Urinary microparticles: HMGB1-containing microparticle [144] Annexin V^+^podocalyxin^+^-micropaticle [145] Ceruloplasmin-containing microparticle [146] Urinary exosomal miRs: miR-29c for early renal fibrosis [147] miR-21, miR-29c, miR-150 for early renal fibrosis [148] miR-21↓ and let-7c↓ for LN disease flare [149] miR-135p for treatment responder patients [150] miR-146a for disease flare [151] miR-195-5p as disease biomarker [152]
Urinary exosomal tRF3-Ile-AAT-1 and tiRNA5-Lys-CTT-1 tsRNA predictor for LN [153]

↓: Decrease in production.

**Table 6 ijms-24-10066-t006:** The potential applications of ncRNAs in treating LN and other novel strategies for LN therapy in the future.

I	Targeting JAK/STAT Signaling Pathways
	• IL-35-related JAK/STAT signaling [166]
	• IFN-γ-induced JAK/STAT signaling and CXCL10 expression [167]
II	Modulation of complement levels [169] or complement factor B level [170]
III	Targeting proinflammatory cytokines or growth factors by microRNAs
	• Suppression of IL-6 expression by miR-410 [189]
	• Suppression of IL-17 and TGF-β expression by miR-125a-3p [192]
	• Suppression of IFN-I and JAK1 signaling by hsa-miR-127-3p [194]
	• Suppression of CSF-1 expression by miR-145 [195,196,197]
	• Decreased M1/M2 polarization (ratio) by locked nucleic acid—anti-miR-150, via SOCS1/JAK1/STAT1 signaling [206,207]
IV	Application of lncRNA and circRNA in the treatment of LN
	• Suppression of IFN-induced signaling by knockdown of lncRNA RP11-2B6.2 [199]
	• Suppression of miR-27b-3p/STING/IRF3/IFN-I signaling by circELK_4_ [200]

## Data Availability

Not applicable.

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
