# Peer review of "Decipher the Immunopathological Mechanisms and Set Up Potential Therapeutic Strategies for Patients with Lupus Nephritis"

_ijms, 2023, doi:10.3390/ijms241210066_

Round 1

Reviewer 1 Report

This review highlights the advance of studies on lupus nephritis. This review is well written and this paper seems to be of interest. However, the reviewer has the following concerns:

It is unclear as to what articles regarding lupus nephritis were included and what was excluded. For instance, how were the articles selected to include? The reviewer thinks that it is better to show the criteria for readers’ understanding.

The title should be re-elaborated to make the article more appealing.

Author Response

Comment 1: This review highlights the advance of studies on lupus nephritis. This review is well written and this paper seems to be of interest.

 Answer: Many thanks for this inspire comment. We will continuously make efforts on this field.

Comment 2: However, the reviewer has the following concerns: It is unclear as to what articles regarding lupus nephritis were included and what was excluded. For instance, how were the articles selected to include? The reviewer thinks that it is better to show the criteria for readers understanding.

Answer: We are extremely sorry for bothering both Reviewers the selective criteria for the articles as references in the present review. We firstly selected the characteristic review articles related to “lupus nephritis” from different viewpoints. Then, we followed the cited research papers supporting their view-point. In addition, the recently published research papers related to the topic are also included by us. Since too many issues are included in this review, some characteristic references would be missed. We apologize for the carelessness. If the Reviewers can recommend or find any missed important literatures in this field, please tell us and we will add them without any hesitation to let the review more comprehension.

Comment 3: The title should be re-elaborated to make the article more appealing.

Answer: Thanks for this valuable suggestion. In the revised version, we have changed the title to “Deciphering the immunopathological mechanisms and setting up potential therapeutic strategies for patients with lupus nephritis”. We hope it will be more appealing and attractive for the readers.

Reviewer 2 Report

In this manuscript, the authors have reviewed the literature on lupus nephritis (LN) with special attention to the pathophysiological mechanisms, the current useful biomarkers in kidney tissues, and potential novel therapeutic strategies for LN. The manuscript is well written and structured and, therefore, it is easy to follow, despite the large number of issues addressed.

It helps the profusion of tables and figures, which summarize the major findings commented by the authors in each subsection.

There are excellent reviews on LN. In a PubMed search performed by the reviewer there were 58 reviews published, only just in 2023, which makes almost impossible to objectively judge what is new in the present review. It would help if the authors make that effort for the readers and put in context this review with those previously published.

Regarding the biomarkers section, only 2 urine proteomics studies are mentioned, which clearly is underestimating the progress in this field. In particular, in relationship with exosomes/extracellular vesicles released via urine.

In line 122, bioinformatics is more correct than bioinformatic

Author Response

Comment 1: In this manuscript, the authors have reviewed the literature on lupus nephritis (LN) with special attention to the pathophysiological mechanisms, the current useful biomarkers in kidney tissues, and potential novel therapeutic strategies for LN. The manuscript is well written and structured and, therefore, it is easy to follow, despite the large number of issues addressed. It helps the profusion of tables and figures, which summarize the major findings commented by the authors in each subsection.

Answer: Thanks for these positive and informative feedbacks. We will keep on it.

Comment 2: There are excellent reviews on LN. In a PubMed search performed by the reviewer there were 58 reviews published, only just in 2023, which makes almost impossible to objectively judge what is new in the present review. It would help if the authors make that effort for the readers and put in context this review with those previously published.

Answer: Thanks for the same comment as pointed-out by the Reviewer [1]. The review articles we selected are quite characteristic in the field of “lupus nephritis”. We also tried to skip the redundant review articles in the beginning. We also tried our best to explain why we selected the review article everywhere in the revised version showing “…. as reviewed by some authors et al.”. We sincerely welcome Reviewers can provide the excellent review or research articles for us to improve the quality of the article.

Comment 3: Regarding the biomarkers section, only 2 urine proteomics studies are mentioned, which clearly is underestimating the progress in this field. In particular, in relationship with exosomes/extracellular vesicles released via urine.

Answer: Thanks again for this constructive suggestion. We have to apologize for only reviewing 2 urinary proteins, 2 immune-related particular cells, 3 cytokines and 2 exosomal tsRNAs as urinary biomarkers in lupus nephritis in the old Table 5. In the revised version, we have added many unique urinary microparticles and exosomal miRs as disease biomarkers in the new Table 5. A paragraph to explain these new urinary biomarkers with their clinical correlations is added in P.23 as shown below.

 Recently, analysis of urinary microparticle components as urinary biomarkers of LN have been reported. Burbano et al [146] found HMGB1-containing microvesicle in urine could become the hallmark of patients with LN. Lu et al. [147] unveiled that the increased podocyte-derived annexinV(+) podocalyxin (+) microparticles are associated with disease activity and renal injury in LN. Gudehithlu et al. [148] noted that increased urinary exosome-derived ceruloplasmin is not only an early biomarker of diabetic kidney but many non-diabetic chronic kidney diseases including LN prior to proteinuria. More interesting, many different urinary exosome-derived miRs have been found as useful biomarkers in patients with LN. Sole et al. [149, 150] reported that urinary exosomal miR-21, miR-29c and miR-150 could become predictors of early renal fibrosis in LN. Tangtanatakul et al. [151] unveiled that down-regulated miR-21 and let-7a in urine exosomes of LN patients could be used to guide the clinical staging during disease flare-up. Garcia-Vives et al. [152] demonstrated that miR-135b exhibited the best predictive value to discriminate responder from non-responder patients. Perez-Hernandez et al. [153] further discovered that urinary exosomal miR-146a was not only a biomarker of albumonemia but correlation with lupus activity, proteinuria, and histological features to discriminate LN patients, but a good baseline marker for SLE flare. Cheng et al. [154] by using a bioinformatic screen combined clinical validation strategy for rapidly mining the exosomal miRs for LN diagnosis and management. The group screened out differntially expressed miRs and mRNAs in LN database for performing a miR-mRNA integrated analysis for selecting out reliable miRs in LN tissues. The authors suggest that urinary exosomal miR-195-5p could be used as a novel biomarker in LN. Besides, miR-195-5p—CXCL10 axis is suggested to become a therapeutic target of LN. Most recently, Chen et al. [155] detected tRNA derived small noncoding RNA (tsRNA) in urine by exosomic methods and showed that upregulated tRF-Ile-AAT-1 and tiRNA5-Lys-CTT-1 tsRNA were present in SLE patients with LN but not in those without LN. These results may suggest that urine-derived different proteins and microparticle-derived nucleic acids can serve as non-invasive biomarkers for the diagnosis and prediction of SLE patients with LN.

                   The new Table 5 is changed to:

Table 5. Potential urinary biomarkers in the patients with LN.

․Urinary BAFF, APRIL and their cognate receptors, TACI/BCMA/BR3 mRNA [122]

․Urinary neutrophils gelatinase-associated lipocalin [138-141]

․Tamm-Horsfall protein in tubulointerstitial damage [142]

․IL-22-binding protein [143]

․CD38+ CD69+ memory T cell and HLA-DR+ CD11c+ macrophage in urinary sediments [144]

․IL-16, CD163 and TGF-b [145]

․Urinary microparticles:

     ․HMGB1-containing microparticle [146]

     ․Annexin V+podocalyxin+-micropaticle [147]

     ․Ceruloplasmin-containing microparticle [148]

․Urinary exosomal miRs:

․miR-29c for early renal fibrosis [149]

․miR-21, miR-29c, miR-150 for early renal fibrosis [150]

         ․miR-21↓and  let-7c↓for LN disease flare [151]

         â€¤miR-135p for treatment responder patient [152]

         â€¤miR-146a for disease flare [153]

․miR-195-5p as disease biomarker [154]

․Urinary exosomal tRF3-Ile-AAT-1 and tiRNA5-Lys-CTT-1 tsRNA predictor for LN [155]

         Besides, 9 new references (Ref.146-154) are added for more completeness.

Comment 4: Comments on the Quality of English Language. In line 122,

 bioinformatics is more correct than bioinformatics.

Answer: It is our fault for this typing-error. We have corrected it in the revised version.